# Novel Model Based on Artificial Neural Networks to Predict Short-Term Temperature Evolution in Museum Environment

**DOI:** 10.3390/s22020615

**Published:** 2022-01-13

**Authors:** Alessandro Bile, Hamed Tari, Andreas Grinde, Francesca Frasca, Anna Maria Siani, Eugenio Fazio

**Affiliations:** 1Department of Fundamental and Applied Sciences for Engineering, Sapienza Università di Roma, via A. Scarpa 16, 00161 Roma, Italy; alessandro.bile@uniroma1.it (A.B.); hamed.tari@uniroma1.it (H.T.); 2Royal Danish Collections, Øster Voldgade 4A, 1355 Copenhagen, Denmark; ag@kosa.dk; 3Department of Physics, Sapienza Università di Roma, P.le A. Moro 5, 00185 Rome, Italy; f.frasca@uniroma1.it (F.F.); annamaria.siani@uniroma1.it (A.M.S.)

**Keywords:** cultural heritage preservation, artificial neural networks, nonlinear autoregressive neural networks, NAR, NARX, forecasting, time series

## Abstract

The environmental microclimatic characteristics are often subject to fluctuations of considerable importance, which can cause irreparable damage to art works. We explored the applicability of Artificial Intelligence (AI) techniques to the Cultural Heritage area, with the aim of predicting short-term microclimatic values based on data collected at Rosenborg Castle (Copenhagen), housing the Royal Danish Collection. Specifically, this study applied the NAR (Nonlinear Autoregressive) and NARX (Nonlinear Autoregressive with Exogenous) models to the Rosenborg microclimate time series. Even if the two models were applied to small datasets, they have shown a good adaptive capacity predicting short-time future values. This work explores the use of AI in very short forecasting of microclimate variables in museums as a potential tool for decision-support systems to limit the climate-induced damages of artworks within the scope of their preventive conservation. The proposed model could be a useful support tool for the management of the museums.

## 1. Introduction

Data-driven models developed with wide range of machine learning algorithms are largerly used in climate and atmospheric modelling [1] Their use, related to microclimate data in cultural heritage domain, is up to now limited and poorly explored. This is mainly due to the limited availability of long-term series of key environmental variables at the sites where the artifacts are exhibited or stored. Although machine learning algorithms allowed to achieve important results in the processing automation and in the improvement of forecasting techniques, their complexity results into long training periods and into the demand for large available datasets [2,3,4]. When only short time series are available, complex networks tend to memorize processes rather than learn them, then failing to identify sudden variations or trends. However, there are particular recurring networks, optimized for time series, which allow rapid learning and good predictions even short time series [5].

This work aims to study a new methodology of analysis (based on Machine Learning) in the field of cultural heritage. Artificial Intelligence (AI) techniques are generally used on very large datasets, characterized by many independent variables and long temporal extension. Datasets with a small number of variables are not optimal for neural learning because they do not provide enough information to the network to develop recognition patterns. In this case, the network could memorize those trends without actually being able to recognize any variations, leading to incorrect results as soon as the trends deviate from what is memorized. In fact, the most critical aspect of AI is “repetition”: the more the models analyze data, the more they are able to adapt independently and learn. Computers learn from previous processing to produce results and take decisions that are reliable.

In this work, we present and compare the NAR (Nonlinear Autoregressive) and NARX (Nonlinear Autoregressive with Exogenous Input) neural algorithms, for the short-time prediction of the microclimate inside museums. The microclimatic variables are characterized by a strong correlation. Furthermore, problems due to sampling often lead to short and fragmented historical series [6]. NAR and NARX models were built on data collected at Rosenborg Castle (hereafter named as RDC data) in Copenhagen (one of the museum partner of the European CollectionCare project); outdoor climate data are extracted from Copernicus Climate Change Service (C3S) Climate Data Store (CDS) [7]. The paper is organized as follows. Rosenborg Castle is presented in Section 2 of the manuscript. Section 3 deals with the description of the collected microclimate data and describes the pre-processing procedures, in order to obtain datasets ready for the learning operations. Section 4 focuses on the description of the structure of the implemented artificial intelligence algorithms and on their characterization. Section 5 introduces the results obtained, proposes an accurate analysis and offers numerous points of reflection on possible future developments.

## 2. Rosenborg Castle

Rosenborg Castle, located in the centre of Copenhagen (Denmark), is one of the museums involved in the CollectionCare project, housing the valuable Royal Danish Collection spanning from the late 16th century of Christian IV to the 19th century. The collection consists of artistic artefacts, paintings and tapestries illustrating the culture and art of the Danish kings from the 17th to the 19th century. A detail of the building is shown Figure 1.

Rosenborg Castle is an old building, not built with modern energy saving systems. The castle has wooden window frames with a single pane of glass, that do not insulate well. The front door is open to the outside environment all opening hours, all year. As well as several open windows on all floors during summer. Therefore, the internal climate is strongly affected by the external one.

A measuring system for hygrothermal variables has been operating in most rooms since the pre-project period. The monitoring system consists of 14 thermo-hygrometers (hereafter named as museum data logger) deployed in twelve rooms. The period under study covers the years from 2013 to 2018, so that six-year indoor temperature (T) and relative humidity (RH) observations were collected with acquisition and recording time within one hour.

By exploiting these time series of data, in this study a forecasting model of the evolution of the museum’s temperature will be created. Such models might help museums’ administrations to optimize the use of heating and cooling systems, in order to reduce management costs and protect the enviroment.

## 3. Microclimate Data in Rosenborg Castle

RDC datasets are characterized by only two variables, temperature (T) and relative humidity (RH). Generally, machine learning models prefer to work on a greater number of independent variables to be crossed together to learn specific behaviors. In fact, as mentioned above, neural algorithms working on limited datasets tend to memorize processes rather than learn them. In order to enlarge the available dataset of the Rosenborg Castle and improve the predective ability of the model, we increasead the number of variables by including outdoor data extracted from Copernicus Climate Change Service (C3S) Climate Data Store (CDS) [7], hereafter named Copernicus data. In fact, as previously highlighted, we expect a strong external influence on the internal climate. The data set is then composed by
RDC air temperature and relative humidity (hereafter named as RDC data);Copernicus data of air temperature, relative humidity, total precipitation, net solar radiation, horizontal wind components (zonal (u) and meridional (v) components).

Before applying NAR and NARX methods, RDC series were undergone to a quality data control. Two quality indices are introduced, that define the completeness (*CoI*) and the continuity (*CI*) of the time series [8]. *CoI* index is defined as the ratio between the number of taken measurements $N_{r}$ and the total number of sampling intervals $N_{t}$:(1)$$CoI=\frac{N_{r}}{N_{t}}$$
where *CoI*∈[0,1]. If *CoI*→0, the series has a low degree of completeness while *CoI* = 1 means that the series is complete, i.e., no missing data. *CI* index provides the reliability of a series with respect to its fragmentation, and it is related to the number of intervals (*NBI*) with missing data compared to the total number of sampling $N_{t}$ [8,9].
(2)$$CI=1-2\cdot \frac{NBI}{N_{t}}$$
where CI∈[0,1]. If CI→0, the series is not continuous while *CI* = 1 means that the series is continuous.

Figure 2a shows the flow chart relating to the data pre-processing operations divided into three phases:the first investigates the level of completeness (*CoI*) and continuity (*CI*) of the data series. Datasets that exceed the threshold value of 0.85 for *CoI* and for *CI*, pass to phase two;the second phase consists in dataset cleaning from NaN;the last phase involves data reorganization into homogeneous sampling matrices,

which involve their reorganization. A well-ordered dataset, without values not readable by the network (NaN) or invalid variations as information, speeds up the learning process by the network and above all improves it. In the following paragraphs, we specifically describe NAR and NARX neural algorithms.

**Figure 2 sensors-22-00615-f002:**
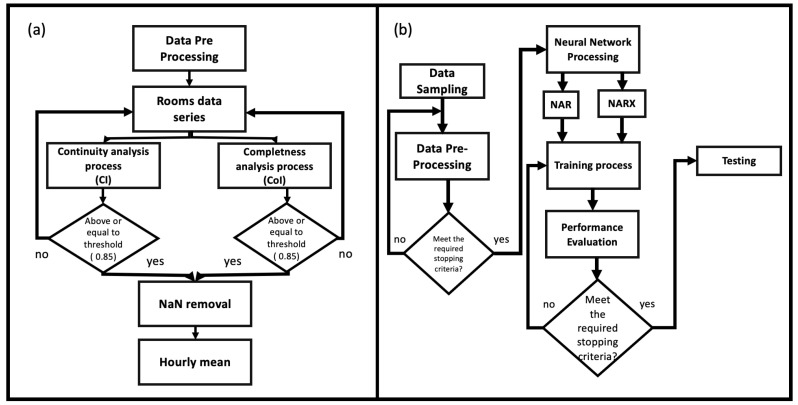
(**a**) Flow Chart of the pre-processing steps: the datasets of each room undergo three levels of selection. First, the completeness (*CoI*) and continuity (*CI*) indices are evaluated; if above a limit threshold they are then cleaned of the presence of NaN and finally standardized in the sampling through a process of mediation of the measurements contained in one hour; (**b**) Flow Chart of the functioning process of the model. Before being processed by the AI, data undergo a pre-processing phase to evaluate the level of information and to be reorganized. Depending on the type of dataset, with a time series or with several time series, the data is fed, respectively, to the NAR or NARX networks. After a training process, which ends with the performance evaluation, the learning is tested (Testing Phase).

The calculation of quality indices to the RDC data reports a high degree of continuity (*CI* > 0.99) for all the exposition halls and a low degree of completeness (average *CoI* < 0.7) but for “The Stone Corridor”, which has a CoI≥0.9 from 6 September 2012 to 3 February 2019 sampling. After analyzing the continuity and completeness of the data, it is necessary to perform two other pre-processing steps. It is therefore essential to eliminate all the NaN present in the dataset and, in the case of several datasets used at the same time, eliminate all the rows corresponding to the row where the NaN (acronym which means Not a Number and indicates the absence of a usable value) of the first dataset is present. After these processing operations, the two datasets have the following characteristics:RDC Stone Corridor dataset: 52,584 rows (from 1 January 2013 00:00:00 to 31 December 2018 00:00:00) and 2 variables: Indoor Temperature ($T_{indoor}$) and Indoor Relative Humidity ($RH_{indoor}$).Copernicus dataset: 52,584 rows (from 1 January 2013 00:00:00 to 31 December 2018 00:00:00) and 6 variables: Outdoor Temperature ($T_{outdoor}$), Outdoor Relative Humidity ($RH_{outdoor}$), Total Precipitation($tp_{outdoor}$), Net solar radiation at the surface ($ssr_{outdoor}$), u wind component (e.g., zonal component, $wind_{u_{outdoor}}$), v wind component (e.g., meridional component, $wind_{v_{outdoor}}$). Data are extracted from gridded observational dataset (spatial resolution of $0.1^{\circ }\times 0.1^{\circ }$) via the Climate Data Store (CDS) infrastructure.

## 4. Artificial Neural Network

The specific neural networks used in this paper are of the Feedforward type (FFNs), for which the information flows in one direction only (forward) through the proces-sing units, called neurons, which are organized in following layers [10,11].

The functional scheme of the entire algorithm is shown in Figure 2b. After the pre-processing operations, the dataset was divided into three different subsets used for training/testing/validation purposes, as shown in Figure 3.

### 4.1. NAR: Nonlinear Autoregressive Network

Most of museums carry out monitoring of a few microclimatic variables, generally temperature and relative humidity. Using datasets limited in extension and in the number of variables, artificial intelligence algorithms tend to overfit the problem:the system gets so used to those data that instead of “understanding” it “memorizes” them. Therefore, any rapid variation may not be recognized. The acronym NAR describes an autoregressive nonlinear neural network, whose output $\bar{y}\left(t\right)$ depends on the previous values of itself [12] through Equation (Equation 3) [13]:(3)$$\bar{y}\left(t\right)=f\left(y\left(t-1\right),y\left(t-2\right),\ldots ,y\left(t-d\right)\right)$$
where *f* is a neuron activation function, $\bar{y}$ is the predicted value at time *t* based on the previous values of the series, whose steps are represented by *d*, and ϵ(*t*) is the approximation error at the time *t*. Hidden layers are characterized by sigmoid activation function while the output layer is characterized by linear activation function.

The optimization for the number of neurons in the hidden layers and time delays were obatined using a trial-and-error procedure [13]. To train the model we have chosen, also in this case, the Levenberg–Marquardt Algorithm (LMA), which requires more memory but it is faster [14,15]. The LMA is used in many applications for solving generic curve-fitting problems but, as with many fitting algorithms, the LMA finds only a local minimum, which is not necessarily the global minimum. For this reason it is necessary to conduct many analyses in order to check whether it is a local minimum (problem not perfectly optimized) or a global minimum (problem perfectly optimized). As shown in Figure 4a, we have implemented a NAR Neural Network characterized by three blocks: input layer, hidden layer and output layer. The same procedure (trial-and-error) was used to establish the time delays. Figure 4a shows the NAR architecture in an open-loop configuration where output is not sent back as input. This configuration is best for the training phase [16]. The first phase is training the network:the output value is known and is used as input to set the internal weights of the network (initially set randomly). The second stage is the test. The calculated output is sent back into the network, and is used, in parallel structure, to estimate subsequent values. The weights are set in order to minimize errors. Once the network has been trained, evaluated and tested, it is possible to make predictions on new data. In case of multiple predictions, the closed-loop configuration, as shown in Figure 4b, seems to give more reliable results [17]. In this configuration each new output (at time *t*) is used together with the previous inputs to predict the next value (at time *t* + 1). Therefore, the open-loop configuration is implemented during network training, while for future predictions we switch to the closed-loop configuration. Our NAR neural network has one input, the indoor temperature, and it returns the prediction of the indoor temperature at a certain time ahead, in this case five future time step. In this model, we have implemented a logistic sigmoid function for the hidden layer and a linear activation function for the output layer, as shown in Figure 4. To find the best performance of the NAR network, an error analysis was conducted. It consists in varying the parameters of the network until the error is minimized. The conducted error analysis showed that the best performances of the NAR network trained with the Levenberg-Maquardt algorithm and on the RDC data is characterized by 50 neurons (nodes) in the hidden layer and 50 time delays, which from now on will be called NAR model 50-50.

### 4.2. NARX: Nonlinear Autoregressive Network with Exogenous Inputs

Nonlinear autoregressive network with exogenous inputs (NARX) is characterized by feedback connections enclosing several layers. The dependent output value y(t+1) is regressed on both its previous quantities and on those of one or more independent exogenous variables [18], provided as inputs. NARX model is based on the following equation [19]:(4)$$\bar{y}\left(t\right)=f\left(y\left(t-1\right),y\left(t-2\right),\ldots ,y\left(t-n_{y}\right),z\left(t-1\right),z\left(t-2\right),z\left(t-n_{z}\right)\right)$$
where $\bar{y}$ is the output value and z represents the input value, $n_{y}$ and $n_{z}$ are the output and the input layers, and the *f* function represents the neuron activation function. The NARX basic architecture is organized similarly to the NAR one and is made up, as shown in Figure 5a, by three main blocks: input, hidden and output layers. In this case too, the training phase must be carried out in an open-loop configuration (Figure 5a). The output information is not used to train the whole network. By modifying the structure into closed-loop configuration (Figure 5b) it is possible to provide multistep predictions.

In NARX networks, the estimated output is fed back to the feedforward network input to return to processing. During the training phase the true output is available and it is possible to supply the real output as input to the network. There are two advantages: first, a cleaner feed-forward network input is obtained; secondly, the neural architecture becomes purely feedforward and only the training phase exploits a static backpropagation [18]. The analysis here proposed has the following characteristics: the target time series is the indoor temperature series $T_{indoor}\left(t\right)$ while the exongenous series are reported below:$x_{1}\left(t\right)=RH_{indoor}\left(t\right)$, indoor relative humidity.$x_{2}\left(t\right)=T_{outdoor}\left(t\right)$, temperature of the external environment.$x_{3}\left(t\right)=RH_{outdoor}\left(t\right)$, relative humidity of the external environment.$x_{4}\left(t\right)=tp_{outdoor}\left(t\right)$, total precipitation of the external environment.$x_{5}\left(t\right)=ssr_{direction_{outdoor}}\left(t\right)$, net solar radiation at the surface of the external environment.$x_{6}\left(t\right)=wind_{u_{outdoor}}\left(t\right)$, zonal wind component (u).$x_{7}\left(t\right)=wind_{v_{outdoor}}\left(t\right)$, zonal wind component (v).

The target series y(t) and the exogenous series $x_{1}\left(t\right)$ are part RDC dataset, while $x_{i}\left(t\right)$ (with *i* = 2, 3, 4, 5, 6, 7) series are retrieved from *Copernicus* dataset. Data are chosen by selecting the sampling period of RDC data.

#### Training Algorithm and Performance Evaluation Metrics

The chosen training algorithm is the Levenberg–Marquardt Algorithm (LMA) [19,20] for both NAR and NARX because of its typical fast convergence, despite of high memory. LM is a back propagation-type algorithm and it is based on modified Gauss–Newton method: basically, it finds the function minima over a space of parameters optimizing the solutions by means of an approximation of the Hessian Matrix [21], determining the variation Δw:(5)$$\Delta w={\left[J^{T}\left(w\right)J\left(w\right)+\lambda I\right]}^{-1}J^{T}\left(w\right)e\left(w\right)$$

In Equation (Equation 5), we can recognize the learning matrix *I*, the vector of network errors e(w) and the learning parameter λ, which is usually set to a randomic value at the beginning and then is updated to minimize error.

J is a Jacobian matrix, $J^{T}$ its transpose, $J^{T}J$ the Hessian matrix and finally *w* represents the neuron’s weight. To evaluate the prediction performance, two criteria were used, Mean Squared Errors (*MSE*) and coefficient of determination ($R^{2}$). Both methods introduce a metric to evaluate the error and therefore how much the predicted measure differs from the real data. *MSE* is a method to evaluate the predicted values $\hat{y}$ with respect to the measured one *y* based on Equation (Equation 6):(6)$$MSE=\frac{1}{n}\sum_{k=i}^{n}{\left({\hat{y}}_{k}-y_{k}\right)}^{2}$$
where *k* presents the actual value and *n* is the total number of values.

The ideal *MSE* should not correspond to zero because this would mean a model that is too tied to the specific dataset and not able to adapt to a general context. If this happens, the model would be overfitted and unable to adapt to data other than those on which it was trained. At the same time, however, very large errors should be avoided because the system underestimates the predictions that would result inadequate. The best condition is therefore in the balance between overfitting (too low *MSE* for training) and underfitting (too high *MSE* for test and validation).

$R^{2}$ method evaluates the predicted values $\hat{y}$ compared to those measured *y* by performing a linear regression and therefore considering the best fit:(7)$$R^{2}=1-\frac{\sum_{k=i}^{n}{\left({\hat{y}}_{k}-y_{k}\right)}^{2}}{\sum_{k=i}^{n}{\left({\bar{y}}_{k}-y_{k}\right)}^{2}}$$
where $\bar{y}$ stands for the average of measured values and *n* for the number of values.

## 5. Results and Discussion

### 5.1. Results with NAR Neural Network

When training a neural network, it is important to analyze the relationship between the forecast errors and the time steps, called time lags [20]. Such relationship is pointed out calculating the autocorrelation $errcorr_{t}$ between errors at different time lags.
(8)$$errcorr_{t}=\sum_{k=-M}^{M}err_{t}err_{t-k}$$
where *M* is the time lag extension of the correlation. The error autocorrelation is shown in Figure 6. At lag zero the model gives no-prediction and consequently the forecast error is the highest: consequently, it is highly correlated with all the other errors too. Moving away from lag zero, the error correlation drops down: the NAR model shows error autocorrelations within the confidence limits *CL*
(9)$$CL=\pm \frac{2errcorr\left(M+1\right)}{\sqrt{L(ei)}}$$
where *errcorr* is the error correlation function, *L* is a function to compute sequence length, and *ei* is the considered error sequence. NAR model shows autocorrelation errors within the 95% confidence limits. The low values founded for the autocorrelation mean that the model is able to escape from errors, performing good future forecasts even with larger errors at previous time lags. Thus, we can conclude the model is suitable because errors are indeed uncorrelated.

The efficiency of our network is then assessed through the *MSE* and the $R^{2}$. These two parameters are calculated and analyzed in the training, testing and validation phases. Figure 7 shows *MSE* versus the number of iterations (epochs) for the three cases of training (blue line), validation (green line) and testing (red line).

The limited dataset results in a validation error of the order of $10^{-2}$, which is larger than the errors shown by NAR in other applications [22,23,24,25,26]. However, we can consider it acceptable since the error on the real measurement is one order of magnitude higher (∼$10^{-1}$ Celsius degree). To compare the results, we calculated the $R^{2}$ too. Figure 8 shows the scatter diagrams of the measured and predicted temperatures: for both training (Figure 8a) and testing (Figure 8b) phases the $R^{2}$ reaches the 0.99 value.

The results of the temperature predictions in the training, validation and testing phases are shown in Figure 9a, while in Figure 9b the corresponding errors between observations and predictions are reported.

Figure 10 shows in blue the microclimate temperature over 5 consecutive hours not used for learning and therefore unknown, while the relative network forecasts are represented in red. In the absence of a protocol that defines how early it is necessary to know the trend of the microclimate, a five-hour interval was considered a good interval as a reference for very short forecasts.

The measures taken into consideration show a trend that changes abruptly, probably due to sudden events such as the opening of a window. At the third hour of prediction, the network deviates from the real trend but is able to approach the real value again from the next forecast. This means that in the course of the training the network has learned to recognize these situations of sudden changes.

### 5.2. Results with NARX

The main advantage of training a network on many variables lies in the possibility of finding many correlative patterns. This improves the learning of the network with respect to the target problem and consequently the predictions.

For NARX network trained with the Levenberg-Maquardt algorithm, the computed error analysis shows best performances with 28 neurons (nodes) in the hidden layer and 28 time delays. Once the temporal steps has been fixed, the increase in the number of neurons leads to an overfitting of the results, while too small temporal steps do not allow to extrapolate sufficient information. Therefore, the best configuration identifies a balance between the complexity of the model and the degree of information obtainable from a specific dataset. In order to identify the best time delays, we have analyzed the relationship between errors predictions and time steps. Figure 11 shows the autocorrelation errors within the 97% confidence limits except again for the zero lag one: thus we can conclude that the model is suitable. Therefore, also for NARX network, lag zero value contains no prediction and consequently the autocorrelation error is higher than those calculated for predicted values.

As in NAR case, the trends of the *MSE* errors are evaluated in the training, validation and testing phases, shown in Figure 12.

The NARX model achieves a lower MSE than the NAR one. This improvement is probably due to the addition of exogenous sets. Consequently, it is expected that larger datasets will further improve these results. The training of the model on the input data was also evaluated on the basis of $R^{2}$. Figure 13a,b show the regressions related to training (Figure 13a) and testing (Figure 13b) phases, both with $R^{2}$ = 0.99.

Figure 14a shows together the true and predicted temperature trends in the three phases. The obtained data overlap almost perfectly.

Finally, the model prediction ability in the short time (5 h) was evaluated. Figure 15 shows temperature unknown data over 5 successive hours and the relative predictions. The slight improvement achieved by moving from the NAR model to the NARX model is also visible in the accuracy of the predictions. Indeed in this configuration the committed error settles between the largest ∼1.84% and the lowest ∼0.65%. Particular attention is to be paid to the third hour of measurement, when the trend changes totally probably for external reasons. Similarly to the NAR case the prediction deviates but the network is then immediately able to straighten the prediction.

## 6. Conclusions

In this work we used and compared two different neural models specific for time series, NAR and NARX, in order to obtain very short-term predictions of the museum microclimate, i.e., within a few hours. The two predictive networks were optimized for the analysis and forecasting of the time series of the “Stone Corridor” exhibition hall of the Rosenborg Castle in Denmark. In both cases, learning is preceded by a data pre-processing phase, necessary to prepare the dataset. This consists in the analysis of the information content of a series of data through two quality indices, CI and COI, on the basis of which a series is considered analyzable or to be discarded. Further reorganization of the data structure follows. The short range prediction using the NAR model showed an *MSE* of lower order of magnitude than the measurement error (i.e., $10^{-2}$), while the NARX model produced an *MSE* of two orders of magnitude lower (i.e., $10^{-3}$). Currently there is no protocol that defines the time horizon within which it is necessary to know how to predict the variations of microclimatic variables. For this reason we have pushed the forecast on a time range of 5 h: in fact, we considered that this period was adequate to follow the trend of the forecasts and, in an operational context, provided sufficient margin to act in advance on the microclimatic conditions by compensating any expected fluctuations. The study on 5-h sets of unknown data showed that the NAR model predicted an initial error of 2.82%, slightly higher than that of the NARX network (1.84%); however, both models improved and the error decreased by an order of magnitude as the networks progressed in the prediction, dropping to 1.60% and 0.65%, respectively. The results obtained constitute an important starting point for predicting the temperature of museums, and are satisfactory with respect to the objectives to be achieved. In particular, the transition from NAR to NARX shows an improvement, albeit a small one, due to the limited datasets available. We can therefore expect that with larger datasets (number of variables and number of data for single variables) the models will be able to give more and more precise predictions. It is however evident that the NARX model should lead to much better results than NAR which has a tendency to saturate with too large datasets as it only manages one variable at a time.

## Figures and Tables

**Figure 1 sensors-22-00615-f001:**
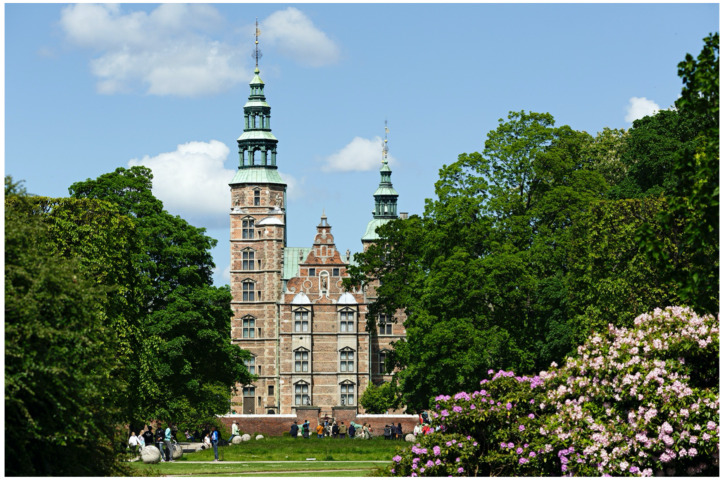
Main facade of Rosenborg Castle, Copenhagen, Denmark.

**Figure 3 sensors-22-00615-f003:**
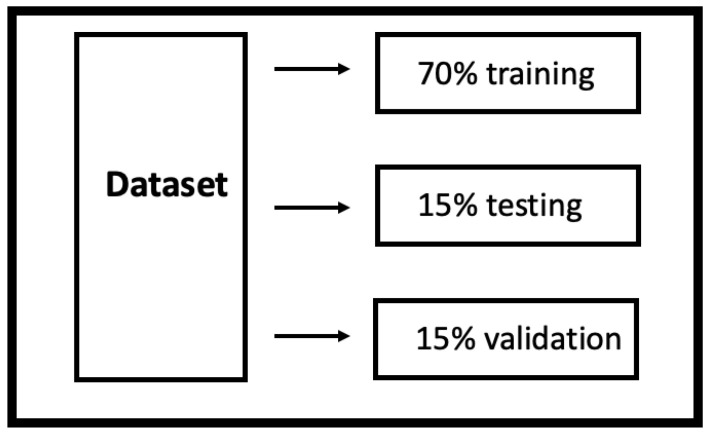
Scheme of the percentage subdivision of the dataset to carry out the training, testing and validation operations.

**Figure 4 sensors-22-00615-f004:**
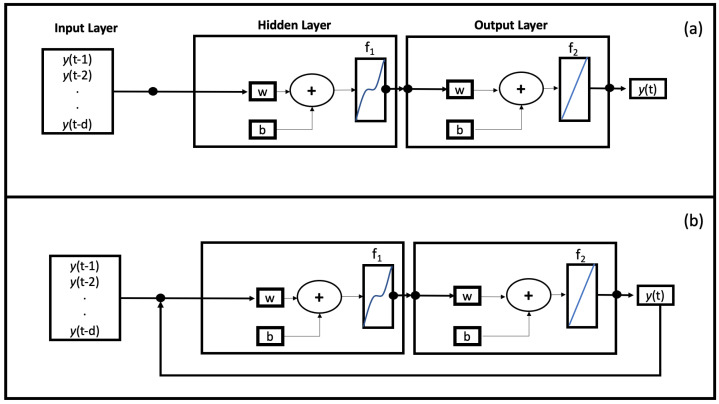
(**a**) Schematic of NAR Architecture Model in open-loop configuration. The first block, Input Layer, represents the data entering the network. The Hidden Layer block represents all neurons, the number of which has been changed during the evaluation phase, which are located between the input and the output. Each neuronal connection is characterized by a weight (w) and a bias (**b**). Hidden Layer and Output Layer are connected through a sigmoid $f_{1}$ activation function. The neurons of the output layer then return an output (predicted data) by activating through the linear function $f_{2}$; (**b**) Schematic of NAR Architecture Model in closed-loop configuration for Multistep Prediction Process.

**Figure 5 sensors-22-00615-f005:**
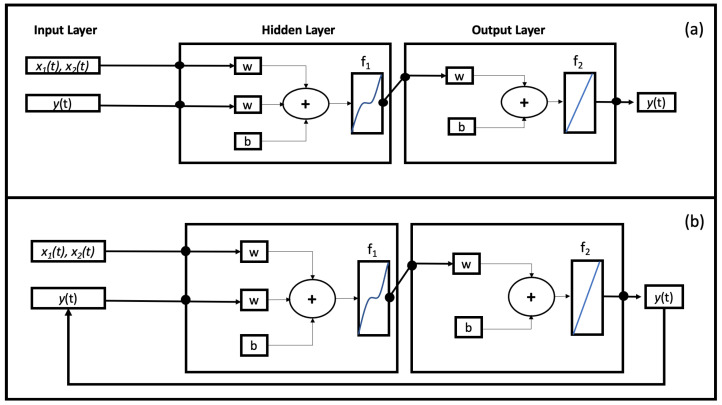
(**a**) Schematic of NARX Architecture Model in open-loop configuration. The first block, Input Layer, represents the data entering the network. The Hidden Layer block represents all neurons, the number of which has been changed during the evaluation phase, which are located between the input and the output. Each neuronal connection is characterized by a weight (w) and a bias (**b**). Hidden Layer and Output Layer are connected through a sigmoid $f_{1}$ activation function. The neurons of the output layer then return an output (predicted data) by activating through the linear function $f_{2}$; (**b**) Schematic of NARX Architecture Model in Closed-Loop configuration for Multistep Prediction process.

**Figure 6 sensors-22-00615-f006:**
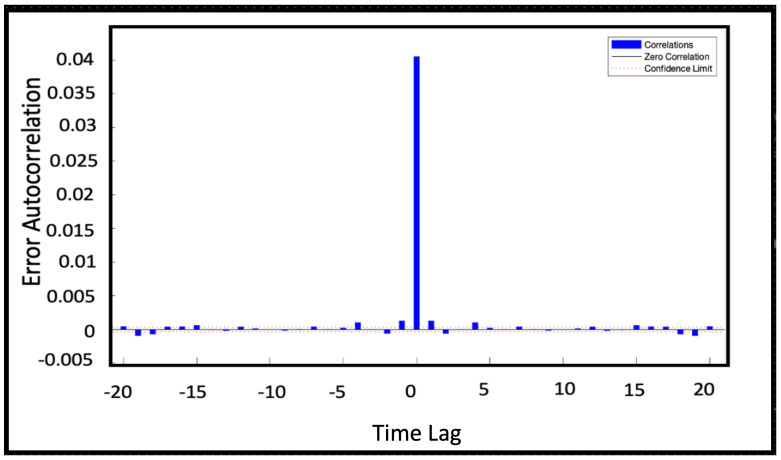
Error autocorrelation of NAR model 50-50. The lag time represents the time gap between values in time series. Tha autocorrelation errors are within the 97% confidence limits (dotted line).

**Figure 7 sensors-22-00615-f007:**
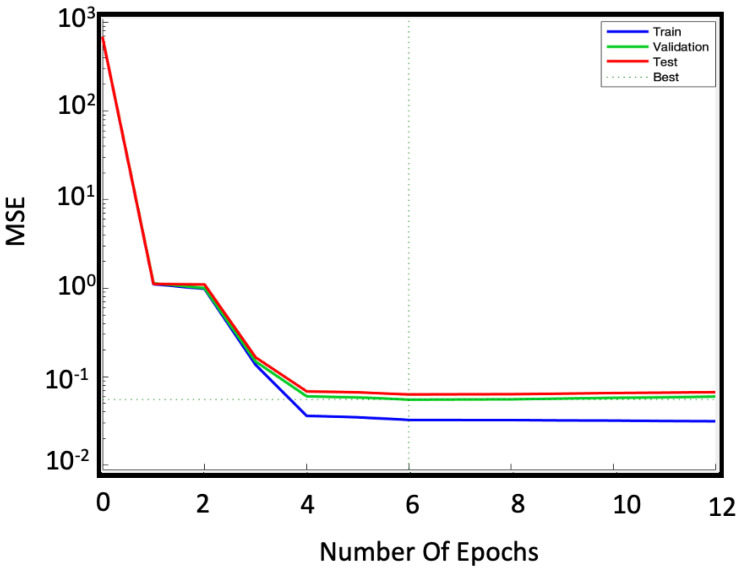
Convergence plot of the NAR model characterized by 50 neurons in the hidden layer and 50 steps of time delay: training phase (blue line), validation phase (green line) and testing phase (red line). It is important to keep these curves under control to avoid overfitting (too low *MSE* during training) and underfitting (*MSE* too high during testing and validation) problems. The optimal *MSE* is identified by the intersection of the dotted lines.

**Figure 8 sensors-22-00615-f008:**
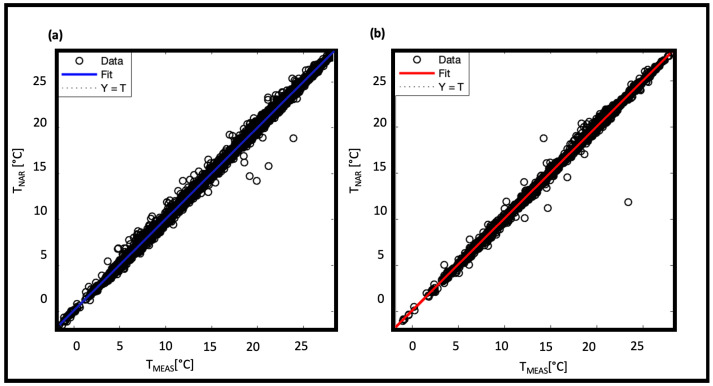
Predicted data regressed on measured data for NAR model 50-50 in training phase and in testing phase.

**Figure 9 sensors-22-00615-f009:**
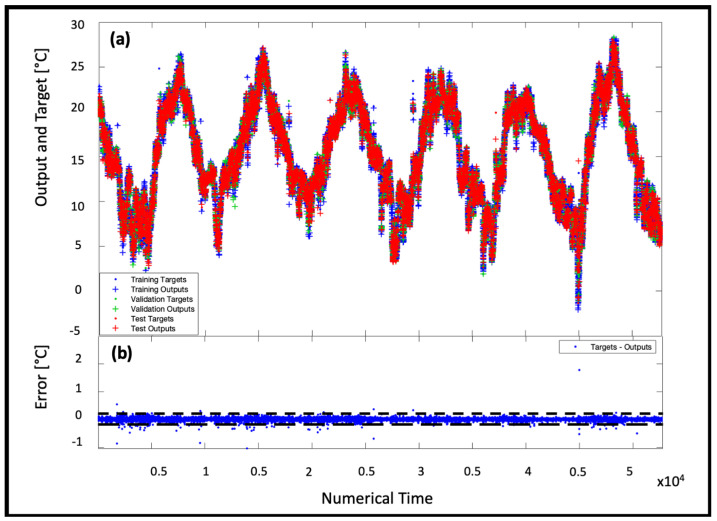
Representation in the NAR 50-50 model (**a**) of the time series response (where the output is the prediction and the target refers to real data) in the three phases of training, validation and testing and (**b**) of the error referred to (**a**) data.

**Figure 10 sensors-22-00615-f010:**
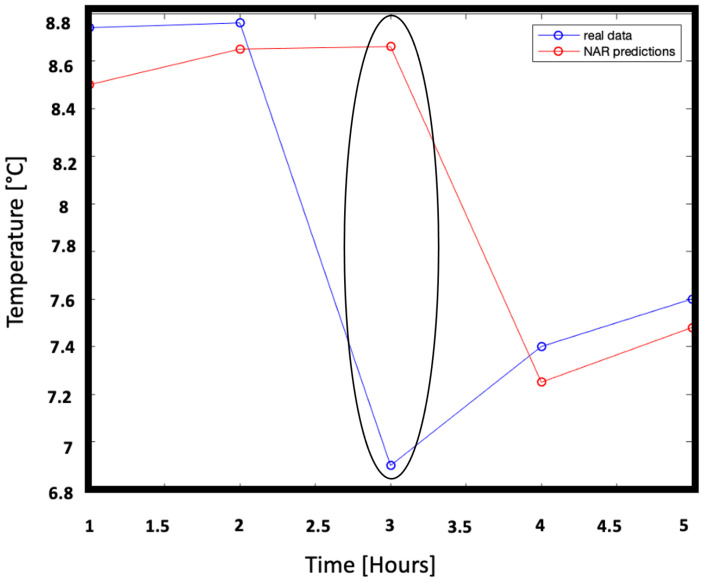
Representation of real data, never fed to the NAR network, and of the respective predictions. The blue line represents the true temperature trend with respect to time (in hours) while the red line represents the predicted temperature trend with respect to time (in hours). The difference between predictions and real data ranges from ∼2.82% to ∼1.60%.

**Figure 11 sensors-22-00615-f011:**
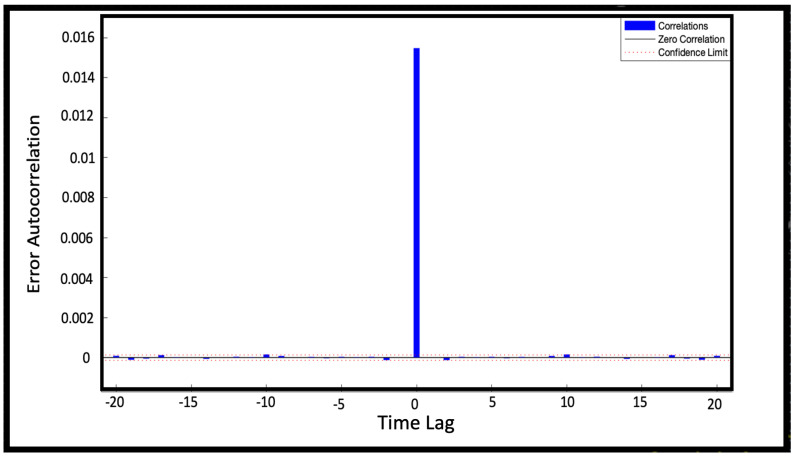
Error autocorrelation of NARX model 28-28. The time lag represents the time gap between values in time series. Image b) represents a zoom of autocorrelation errors. It can be noted that the autocorrelation errors are within the 97% confidence limits (highlighted by the dotted lines).

**Figure 12 sensors-22-00615-f012:**
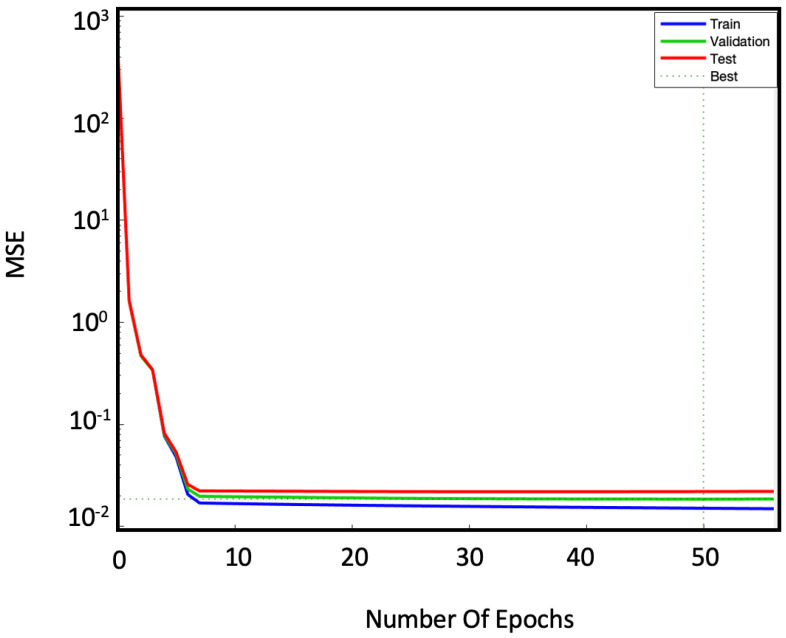
Convergence plot of the NARX model characterized by 28 neurons in the hidden layer and 28 steps of time delay. The blue curve represents the MSE trend for the training phase, the green one for the validation phase and the red one for the testing phase. The dotted lines identify, at their point of intersection, the optimal value of *MSE*. It is important to keep these curves under control to avoid overfitting (too low *MSE* during training) and underfitting (*MSE* too high during testing and validation) problems. The optimal *MSE* is identified by the intersection of the dotted lines.

**Figure 13 sensors-22-00615-f013:**
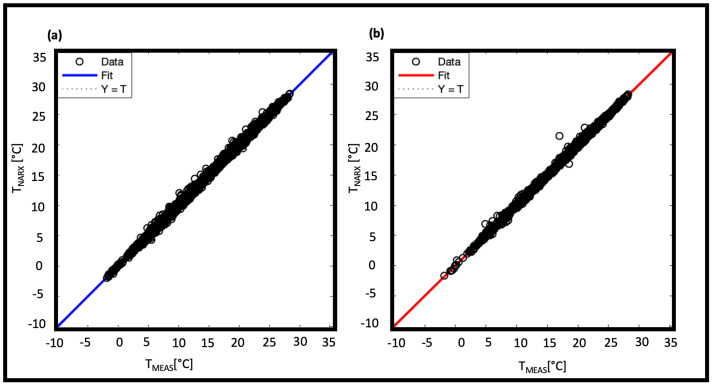
Predicted data regressed on measured data for NARX model 28-28 in training phase and in testing phase.

**Figure 14 sensors-22-00615-f014:**
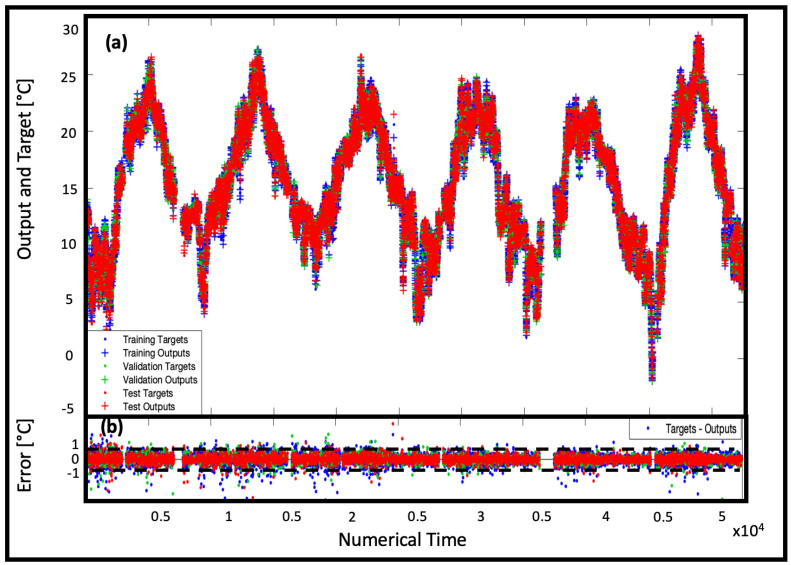
Representation in the NARX 28-28 model (**a**) of the time series response in the three phases of training, validation and testing and (**b**) of the error referred to (**a**) data.

**Figure 15 sensors-22-00615-f015:**
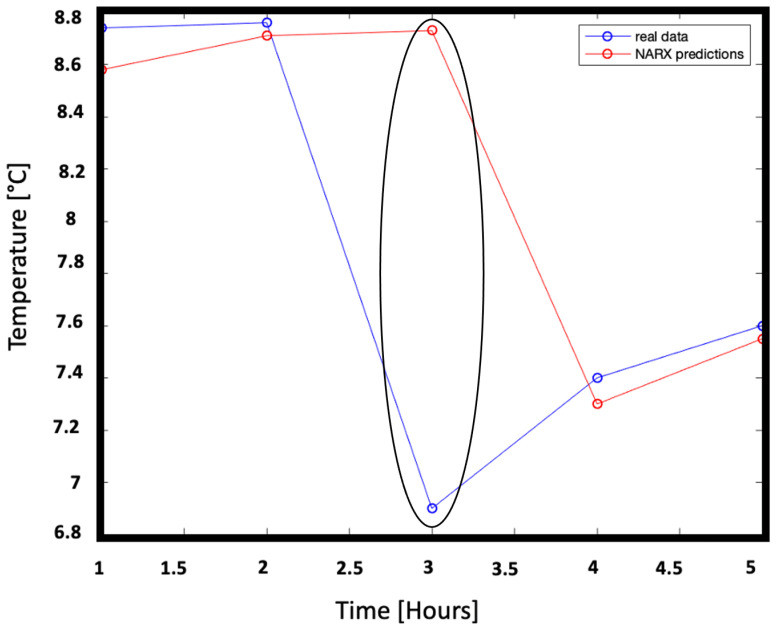
Representation of real data, never fed to the NARX network, and of the respective predictions: measurements (blue line) and predictions (red line). The the difference between predictions and real data ranges from ∼1.84% to ∼0.65%.

## Data Availability

Not applicable here.

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
