# Peer review of "Novel Model Based on Artificial Neural Networks to Predict Short-Term Temperature Evolution in Museum Environment"

_sensors, 2022, doi:10.3390/s22020615_

Round 1

Reviewer 1 Report

This paper is of interest and relevance in the field of preventive conservation, especially in the management of the museum environment.

This paper can be accepted after minor revisión:

- Page 3. Line 70 Adiministrations. The authors must replace this word. It´s not correct. 

- The authors have considered the economic cost of these models?

Author Response

We thank the referee for his/her report. About the questions:

REFEREE #1

The word “Administrations” (line 70) has been corrected.

Economic costs: in fact, it is not clear what economic costs have to be faced. The monitoring of the microclimate requires installation costs of the measurement systems but these cannot be considered here. The model we studied could run on a simple mobile phone app with extremely low costs. Being able to foresee important thermal fluctuations in advance could help museum managers to implement rapid prevention actions to counter them. This could have a cost, but certainly lower than the costs of extraordinary maintenance of the objects on display if damaged by microclimatic conditions that are not their own induced by rapid fluctuations (thermal shock, excessive humidity or excessive dryness).

Reviewer 2 Report

The paper is trending, explores the use of AI to predict short-term temperature evolution in museum environment. The study applied the NAR and NARX models to Rosenborg microclimate time series.

In the introduction section the authors explain why they chosen to use methodology based on Machine Learning and the AI to predicting short-term microclimatic values in museum environment as "repetition".

It was compared the NAR and NARX neural algorithms, for the short time prediction of the microclimate inside museums. The two predictive networks were optimized for the analysis and forecasting of the time series of the "Stone Corridor" exhibition hall of the Rosenborg Castle in Denmark.

The experimental part is clear presented, with adequate results and discussion.

The references are appropriate.

In conclusions ‘section the authors emphasise their original contributions for prediction the museum microclimate within a few hours.

This is a well written paper, useful to read. The subject is well within the scope of the journal, and the paper fulfils all the requirements to be published.

As an advice: you can improve your research by develop a software application for mobile, for example.

Author Response

We thank the referee for his/her report. No questions were posed

Reviewer 3 Report

Dear authors,

The article's subject and methodology are very interesting. I only made some minor suggestions to improve it, listed below.

There are two meanings of the RDC acronym. In line 44, you mention data collected at Rosenborg Castle. However, in lines 57 and 58, it is described as the Royal Danish Collection. Please choose one of them and change the other.

In line 65, you say that the dataset started in 2012. In contrast, lines 118 and 121 display the dataset beginning in 2013. Please fix it.

You have too many figures in your article. In my opinion, figure 4 is not necessary, and figure 2 and 3 could be together in just one flowchart. In addition, figures 5 and 6 could also be joined in just one figure, which could be easy to compare the NAR open and closed-loop configuration. The same could be done in figures 7 and 8 for the NARX open and closed layout.

In addition to these small suggestions, I Have one central question:

It would be right to compare inside and outdoor climate data even though the indoor data usually has some heater or cooling system, which could change the relationship pattern?

Sincerely

Author Response

We thank the referee for his/her report. About the questions:

#1: line 44: we have eliminated the phrase in brackets (hereafter named as RDC data)

#2: Dataset: 2012 o 2013? lines 65-118-121

#3: We have reduced the number of figures by assembling together fgg. 2 and 3, figs. 5 and 6, and figs. 7-8 . We have not deleted fig. 4 (now fig.3), as suggested by the referee, because we believe it make the whole description clearer.

#4: We thank the referee for the final question he asked us. In the meantime, we can say that in the case of Rosemborg Castle the internal climate is strongly affected by the external one: in fact, it is a relatively old building with non-hermetic wooden window frames. Being able to have data from different buildings with different energy classes, it would be interesting to include this information in the model to study the correlation of internal microclimatic fluctuations with weather conditions.

On this topics, we have added two phrases in the text, one at the lines 62-66 and one at the lines 86-87. These phrases describe why the external weather is so important for the internal climate.